# Identifying causal variants by fine mapping across multiple studies

**Nathan LaPierre**[1☯], **Kodi Taraszka**[1☯], **Helen Huang**[2], **Rosemary He**[3], **Farhad Hormozdiari**[6], **Eleazar Eskin**[1,4,5]*

**1** Department of Computer Science, University of California, Los Angeles, California, United States, **2** Bioinformatics Interdepartmental Program, University of California, Los Angeles, California, United States, **3** Department of Mathematics, University of California, Los Angeles, California, United States, **4** Department of Human Genetics, University of California, Los Angeles, California, United States, **5** Department of Computational Medicine, University of California, Los Angeles, California, United States, **6** Harvard T.H. Chan School of Public Health, Boston, Massachusetts, United States

☯ These authors contributed equally to this work.
\* eeskin@cs.ucla.edu

**Editor:** Eleftheria Zeggini, Helmholtz Zentrum München Deutsches Forschungszentrum für Umwelt und Gesundheit: Helmholtz Zentrum Munchen Deutsches Forschungszentrum fur Gesundheit und Umwelt, GERMANY

**Data Availability Statement:** MsCAVIAR is free and open source, and the source code is available

## Abstract

Increasingly large Genome-Wide Association Studies (GWAS) have yielded numerous variants associated with many complex traits, motivating the development of "fine mapping" methods to identify which of the associated variants are causal. Additionally, GWAS of the same trait for different populations are increasingly available, raising the possibility of refining fine mapping results further by leveraging different linkage disequilibrium (LD) structures across studies. Here, we introduce multiple study causal variants identification in associated regions (MsCAVIAR), a method that extends the popular CAVIAR fine mapping framework to a multiple study setting using a random effects model. MsCAVIAR only requires summary statistics and LD as input, accounts for uncertainty in association statistics using a multivariate normal model, allows for multiple causal variants at a locus, and explicitly models the possibility of different SNP effect sizes in different populations. We demonstrate the efficacy of MsCAVIAR in both a simulation study and a trans-ethnic, trans-biobank fine mapping analysis of High Density Lipoprotein (HDL).

## Author summary

Genome-Wide Association Studies (GWAS) have successfully identified numerous genetic variants associated with a variety of complex traits in humans. However, most variants that are associated with traits do not actually cause those traits, but rather are correlated with the truly causal variants through Linkage Disequilibrium (LD). This problem is addressed by so-called "fine mapping" methods, which attempt to prioritize putative causal variants for functional follow-up studies. In this work, we propose a new method, MsCAVIAR, which improves fine mapping performance by leveraging data from multiple studies, such as GWAS of the same trait using individuals with different ethnic backgrounds ("trans-ethnic fine mapping"), while taking into account the possibility that causal variants may affect the trait more or less strongly in different studies. We show in

on GitHub: (https://github.com/nlapier2/MsCAVIAR). Code and instructions to replicate our results are also available on GitHub: (https://github.com/nlapier2/mscaviar_replication). The UK Biobank HDL Cholesterol dataset can be downloaded from https://broad-ukb-sumstats-us-east-1.s3.amazonaws.com/round2/additive-tsvs/30760_raw.gwas.imputed_v3.both_sexes.tsv.bgz. The Biobank Japan HDL Cholesterol dataset can be downloaded by accessing http://jenger.riken.jp/en/result and clicking the "Download" button next to "High-density-lipoprotein cholesterol (HDL-C) (autosome)". The 1000 Genomes data was downloaded by using the following script https://github.com/gkichaev/PAINTOR_V3.0/blob/master/PAINTOR_Utilities/CalcLD_1KG_VCF.py; instructions are available at https://github.com/gkichaev/PAINTOR_V3.0/wiki/2a.-Computing-1000-genomes-LD.

**Funding:** NL would like to acknowledge the support of National Science Foundation grant DGE-1829071 and National Institute of Health grant T32 EB016640. EE is supported by National Science Foundation grants 0513612, 0731455, 0729049, 0916676, 1065276, 1302448, 1320589 and 1331176, and National Institutes of Health grants K25-HL080079, U01-DA024417, P01-HL30568, P01-HL28481, R01-GM083198, R01-ES021801, R01-MH101782, and R01-ES022282. The funders had no role in study design, data collection and analysis, decision to publish, or preparation of the manuscript.

**Competing interests:** The authors have declared that no competing interests exist.

simulations that our method reduces the number of variants needed for functional follow-up testing versus other methods, and we also demonstrate the efficacy of MsCAVIAR in a trans-ethnic, trans-biobank fine mapping analysis of High Density Lipoprotein (HDL).

## Introduction

Genome-Wide Association Studies (GWAS) have successfully identified numerous genetic variants associated with a variety of complex traits in humans [1–3]. However, most of these associated variants are not causal, and are simply in Linkage Disequilibrium (LD) with the true causal variants. Identifying these causal variants is a crucial step towards understanding the genetic architecture of complex traits, but testing all associated variants at each locus using functional studies is cost-prohibitive. This problem is addressed by statistical "fine mapping" methods, which attempt to prioritize a small subset of variants for further testing while accounting for LD structure [4].

The classic approach to fine mapping involves simply selecting a given number of SNPs with the strongest association statistics for follow-up, but this performs sub-optimally because it does not account for LD structure [5]. Bayesian methods that did account for LD structure were developed [6, 7], but were based upon the simplifying assumption that each locus only harbors a single causal variant, which is not true in many cases [8]. Additionally, many early methods required individual-level genetic data, whereas many human GWAS often provide only summary statistics due to privacy concerns. CAVIAR [8] introduced a Bayesian approach that relied only on summary statistics and LD, accounted for uncertainty in association statistics using a multivariate normal (MVN) distribution, and allowed for the possibility of multiple causal SNPs at a locus. This approach was widely adopted and later made more efficient by methods such as CAVIARBF [9], FINEMAP [10], and JAM [11].

There is growing interest in improving fine-mapping by leveraging information from multiple studies. One of the most important examples of this is trans-ethnic fine mapping, which can significantly improve fine mapping power and resolution by leveraging the distinct LD structures in each population [12–14], as seen in methods such as trans-ethnic PAINTOR [15] and MR-MEGA [16]. Intuitively, the set of SNPs that are tightly correlated with the causal SNP(s) will be different in different populations, allowing more SNPs to be filtered out as potential candidates. However, the varying LD patterns also present a unique challenge in the multiple study setting that trans-ethnic fine mapping methods must handle. Additionally, while there is evidence that the same SNPs drive association signals across populations [12, 17, 18], there is also heterogeneity in their effect sizes [13, 17, 18], presenting another challenge. Existing methods either assume a single causal SNP at each locus [16, 19] or do not explicitly model heterogeneity [15], limiting their power [20].

In this paper, we present MsCAVIAR, a novel method that addresses these challenges. We retain the Bayesian MVN framework of CAVIAR while introducing a novel approach to explicitly account for the heterogeneity of effect sizes between studies using a Random Effects (RE) model. Our method requires only summary statistics and LD matrices as input, allows for multiple causal variants at a locus, and models uncertainty in association statistics and between-study heterogeneity. The output is a set of SNPs that, with a user-set confidence threshold (e.g. 95%), contains all causal SNPs at the locus.

We show in simulation studies that MsCAVIAR outperforms existing trans-ethnic fine mapping methods [15] and extensions of methods such as CAVIAR [8] to the multiple study setting. We further demonstrate the efficacy of MsCAVIAR in a trans-ethnic, trans-biobank

fine mapping analysis of High Density Lipoprotein. MsCAVIAR is freely available at https://github.com/nlapier2/MsCAVIAR.

## Results

### MsCAVIAR overview

Our method, MsCAVIAR, takes as input the association statistics (e.g. Z-scores) and linkage disequilibrium (LD) matrix for SNPs at the same locus in each study (Fig 1A). The LD matrix can be computed from in-sample genotyped data or appropriate reference panels such as the 1000 Genomes project [21] or HapMap project [22]. MsCAVIAR computes and outputs a minimal-sized "causal set" of SNPs that, with probability at least $\rho$, contains *all* causal SNPs, and ideally contains far fewer SNPs than the set of significant SNPs obtained via meta-analysis (Fig 1B).

By our definition of a causal set, every causal SNP must be contained in the set with high probability, but not every SNP in the set needs to be causal. Concretely, each SNP can be assigned a binary causal status: 1 for causal or 0 for non-causal. So long as none of the SNPs outside of the causal set are set to 1, the assignments are compatible with our definition of a causal set. We can represent these causal status assignments in a binary vector with one entry for each SNP denoting its causal status; we call such a vector a "configuration" and denote it as $C$. For each configuration $C$ compatible with the causal set, we compute its (posterior) probability in a Bayesian manner: the probability of a configuration of SNPs being causal given the association statistics can be computed by modeling a prior probability for that configuration and a likelihood function for the association statistics given the assumed causal SNPs given by $C$ (see Methods for details).

The overall likelihood function can be decomposed into a product over the likelihood function for each study, since we assume that the studies are independent. More specifically, we assume that there is a true global effect size for a SNP over all possible populations, around which the effect sizes for that SNP in different studies are independently drawn according to a heterogeneity variance parameter (Methods). This allows MsCAVIAR to model the fact that

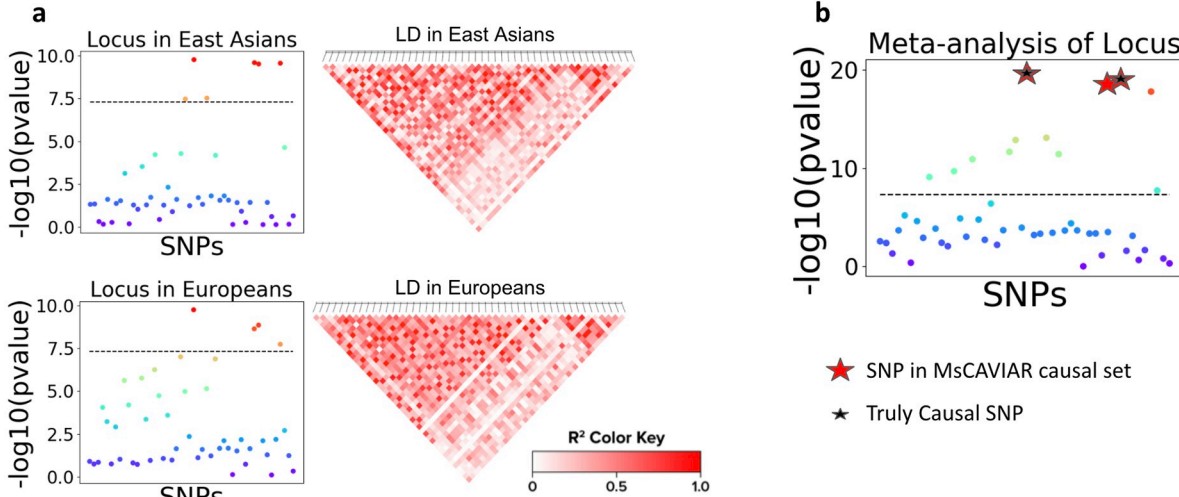

**Fig 1. Overview of MsCAVIAR.** (A) Simulated Z-scores for SNPs at a quantitative trait locus in two different populations: East Asians (top) and Europeans (bottom), shown by their $-log_{10}$(p-value). LD matrices for these populations were derived using data from the 1000 Genomes project. These are the input files to MsCAVIAR. (B) Meta-analysis results for this locus, showing many significant SNPs. Also displayed are the SNPs that are in the causal set that MsCAVIAR returns (red stars) and the truly causal SNPs (black stars).

effect sizes of a SNP across different studies are related, but not equal. Because we expect the summary statistics to be a function of their LD with the causal SNPs, the parameters of the likelihood function for each study are different, assuming the studies have different LD patterns. By computing the product over the likelihood of each study, we are able to account for their different LD patterns to determine the likelihood over all the studies.

The posterior probability for a causal set is then computed by summing the posterior probabilities of all compatible configurations, and then dividing by the sum of the posterior probabilities for all possible configurations. We start by assessing causal sets containing only one SNP, and then causal sets containing two SNPs, and then three SNPs, and so on until a causal set exceeds the posterior probability threshold $\rho$. In practice, $\rho$ is set to a high value such as 95%.

## MsCAVIAR improves fine mapping resolution in a simulation study

We now describe our simulation study to evaluate the performance of MsCAVIAR as compared with other methods. We selected two samples of 9,000 unrelated individuals from the UK Biobank [23], one with European ancestry and the other with Asian ancestry. In order to generate realistic fine mapping scenarios, we centered 100kbp windows around SNPs that reached genome-wide significant association with High-Density Lipoprotein cholesterol in the UK Biobank [23] summary statistics released by the Neale lab [24]. From these windows, we selected three loci that reflected high, medium, and low patterns of LD as defined by the proportion of SNPs with at least 90% LD (32%, 25%, and 8%, respectively). We then obtained the imputed genotype data for these loci for our samples in the UK Biobank. The loci were filtered for missing genotypes ($> 0\%$) and low minor allele frequency ($< 1\%$). The loci with low, medium, and high LD had 144, 126, and 154 SNPs, respectively.

We then simulated causal SNPs and their effect sizes $\beta_i \sim \mathcal{N}(\frac{5.2}{\sqrt{9000}}, 1)$, for the cases of 1, 2, or 3 causal SNPs randomly chosen within each locus. For simplicity, we take the absolute value of the effect size and restrict causal SNPs to being positively correlated with each other. We then used GCTA [25] to simulate phenotypes using different heritability levels: 0.2%, 0.4%, 0.6%, 0.8%, and 1%, times the number of causal SNPs. Concretely, GCTA simulates the phenotypes $y$ according to $y = X\beta + e$, where $X$ is the standardized genotype matrix for the causal variant(s), $\beta$ is the vector of causal variant effect sizes, and $e$ is a vector of environmental noise terms where each $e_i = \sigma_g^2(1/h^2 - 1)$. In other words, the environmental variance is scaled to achieve the desired heritability. Thus, modulating the heritability affects the strength of the association signal between variants and the phenotype, while drawing different $\beta_i$ for different causal variants allows for the modeling of heterogeneity.

Finally, we run a linear regression using fastGWA [26] to generate the summary statistics. We simulated 20 replicates (re-drawing the causal SNPs and their effect sizes) for each level of heritability and number of causal SNPs for a total of 900 simulations.

Using this data, we compared MsCAVIAR to the trans-ethnic mode of PAINTOR [15] and to CAVIAR [8] run on Asians and Europeans, individually (Fig 2). For each number of causal SNPs (1, 2, or 3), we averaged the results across all simulated scenarios. For each method, we provided the in-sample LD and the summary statistics described above. All methods were run with posterior probability threshold $\rho^* = 0.95$, so methods with 95% or higher sensitivity were considered "well-calibrated" (dashed line in Fig 2A). MsCAVIAR's heterogeneity parameter was set to $\tau^2 = 0.52$ (Methods). We also evaluated methods for the size of their returned causal sets (Fig 2B) because, conditioned on having a well-calibrated recall, it is preferable to return a small causal set. This can be thought of as higher "precision", as non-causal SNPs in the causal sets can be thought of as "false positives".

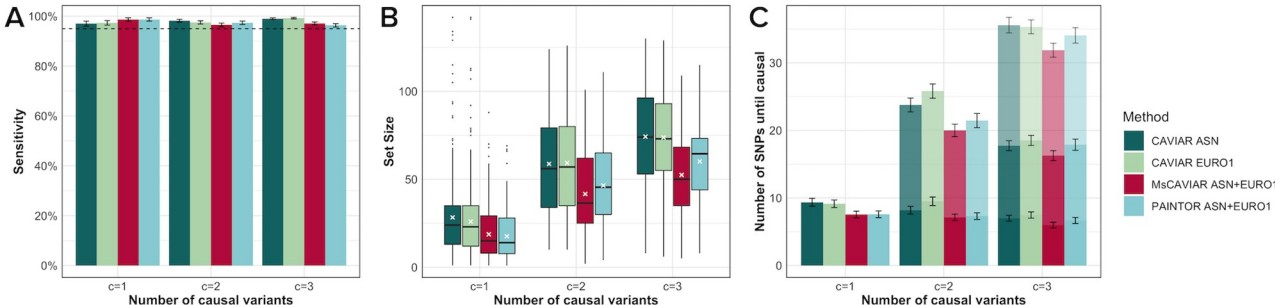

**Fig 2. Comparison of sensitivity, precision, and set sizes using simulated data.** We compare MsCAVIAR, PAINTOR, and CAVIAR with $c \in \{1, 2, 3\}$ causal variants implanted with results averaged over 20 replicates for 3 loci and 5 levels of heritability for all 3 values of $c$. (A) Bar graph indicating the sensitivity of each method with a dashed line to reflect the expected posterior probability, $\rho$, of recovering all causal SNPs (B) Box plots showing the average set sizes returned by the methods. Each box is the interquartile range of causal set sizes with the middle black line representing the median, and the white crosses showing the mean. (C) Bar graph displaying the average the number of SNPs in descending order of posterior inclusion probability (PIP) until 1, 2, or 3 causal SNPs is identified. Stacked bars represent increasing numbers of causal SNPs identified, until the true number of causal SNPs (x-axis) are identified.

All of the methods in this assessment were well-calibrated (Fig 2A), which is expected, as previously shown for CAVIAR [8] and PAINTOR [15]. For each number of causal SNPs, MsCAVIAR and PAINTOR returned substantially smaller set sizes than CAVIAR run on either population individually, highlighting the benefit of utilizing information from multiple studies.

With one causal SNP in the locus, MsCAVIAR and PAINTOR had similar causal set sizes, with MsCAVIAR's mean and median set sizes being 18.7 and 15.0 and PAINTOR's being 17.6 and 14.0, respectively. When there were two causal SNPs simulated, MsCAVIAR's causal sets were smaller on average than PAINTOR's, and the difference increased when three causal SNPs were simulated. When two causal SNPs were simulated, MsCAVIAR's mean and median set sizes were 41.6 and 36.5, respectively, while PAINTOR's mean and median set sizes were 46.4 and 45.5, respectively. Finally, with three causal SNPs, MsCAVIAR had mean and median set sizes of 52.4 and 50.0, respectively, and PAINTOR's were 60.1 and 64.5, respectively.

As the goal of most statistical fine mapping methods is to prioritize variants for functional follow-up, it lends the question of how informative a variant's posterior probability is to its causal status. We, therefore, sort the SNPs in descending order of posterior probability to determine on average how many SNPs are added to the causal set before the causal SNPs are placed in the causal set.

We evaluated this quantity for MsCAVIAR, PAINTOR, and CAVIAR run on the Asian and European populations (Fig 2C). MsCAVIAR and PAINTOR were generally better at prioritizing variants than CAVIAR, again highlighting the importance of utilizing multiple studies when possible. On average, MsCAVIAR was able to capture the causal variant(s) with fewer SNPs than PAINTOR.

## Trans-Biobank fine mapping of high density lipoprotein loci

In order to evaluate the performance of MsCAVIAR on real data, we performed a trans-ethnic, trans-biobank fine mapping analysis of High Density Lipoprotein (HDL) using summary statistics from the UK Biobank (UKB) [23, 24] and Biobank Japan (BBJ) [27, 28] projects. These studies involved 361,194 and 70,657 people, respectively. The UKB summary statistics, obtained from the Neale lab [24], were generated using only White British individuals.

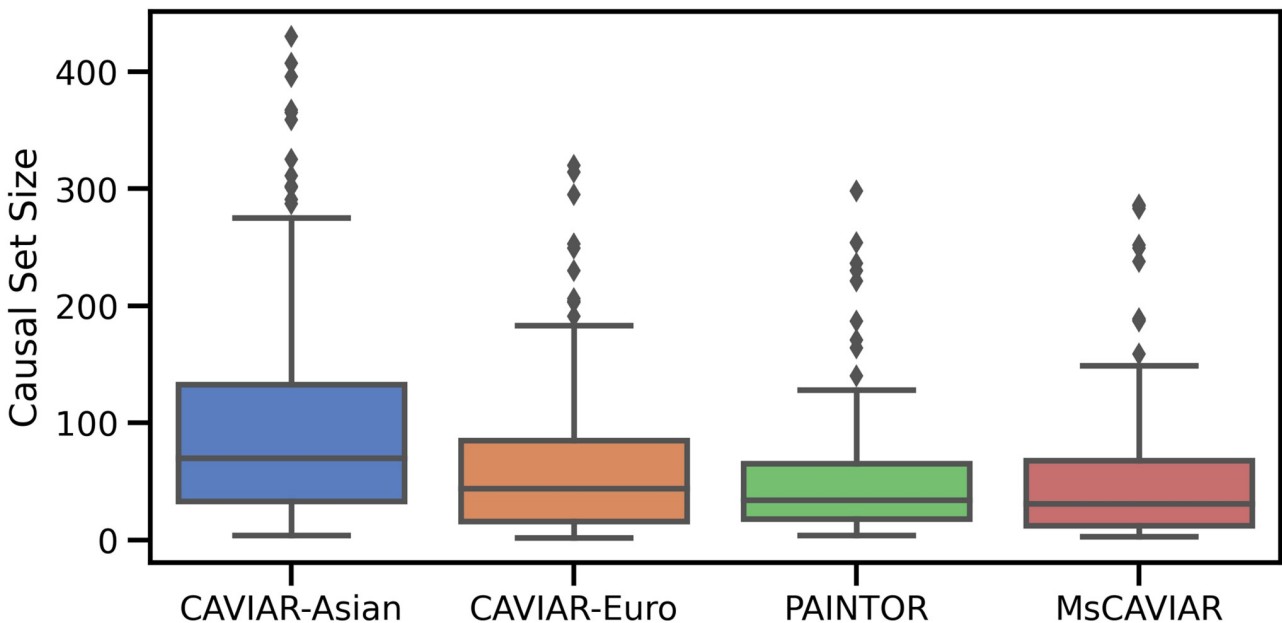

**Fig 3. Comparing fine mapping resolution in trans-ethnic HDL analysis.** Comparison of the results of MsCAVIAR when applied to 185 loci from two high-density lipoprotein (HDL) GWAS, White European people from the UK Biobank [23, 24] and Japanese people from Biobank Japan [27, 28], versus trans-ethnic PAINTOR [15] and applying CAVIAR [8] to each population individually. The y-axis is the size of the causal set for each locus. The boxes represent the interquartile range of causal set sizes identified by each tool, the lines inside the boxes represent the median, and the whiskers extend to the non-outlier extremes. Outliers are represented as dots above or below the whiskers.

To generate loci for fine mapping, we centered 1 megabase windows around genome wide-significant peak SNPs (p-value $\leq 5 * 10^{-8}$), discarding all SNPs that did not reach even marginal significance ($p > 0.05$), as they were highly unlikely to be informative and would slow down analyses. We also excluded all loci with fewer than ten SNPs in each study after filtering SNPs with $p > 0.05$, as fine mapping may not be seen as necessary or may even be trivial for existing methods when there are only a few strongly associated SNPs. Two very large loci were excluded for computational reasons. We excluded loci from chromosome six, where there were numerous statistically significant SNP effect sizes due to the presence of human leukocyte antigen (HLA) regions.

The procedures described above yielded 185 loci consisting of 29,479 SNPs in total. Individual locus sizes ranged from 11 to 755 SNPs. All but two SNPs in the loci had a minor allele frequency of at least 1% at least one of the studies. Linkage disequilibrium (LD) matrices were generated from the 1000 Genomes project [21], with "European" and "East Asian" as the population names, using the "CalcLD_1KG_VCF.py" script from the PAINTOR [29] GitHub repository. We used the 1000 Genome project to generate LD to reflect the common situation where summary statistics are available but not the full genotyped data [8, 10].

We ran CAVIAR [8], the trans-ethnic mode of PAINTOR [15], and MsCAVIAR on these loci, and evaluated their causal set sizes, since these methods have been shown to be well-calibrated and no ground truth is available (Fig 3). For MsCAVIAR, we set the heterogeneity parameter $\tau^2$ (Methods) to its default value of 0.52. For CAVIAR, we evaluated its performance when applying it to only the Asian (BBJ) data or to only the European (UKB) data. For all methods, we set the posterior probability threshold $\rho^*$ to 95% and set the maximum number of causal SNPs to 3.

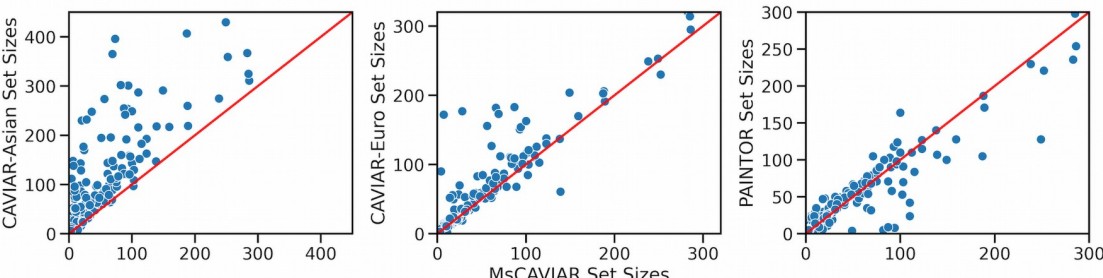

**Fig 4. Comparison of methods' set sizes for each locus in the trans-ethnic HDL analysis.** Comparison of the returned causal set sizes of MsCAVIAR when applied to two high-density lipoprotein (HDL) GWAS, White European people from the UK Biobank [23, 24] and Japanese people from Biobank Japan [27, 28], versus trans-ethnic PAINTOR [15] and applying CAVIAR [8] to each population individually. In each scatter plot, each point reflects a specific locus, and the x-coordinate is MsCAVIAR's returned causal set size, while the y-coordinate is a different method's causal set size. Diagonal lines representing equal set sizes were plotted for each scatter plot. Points above the line represent loci where the alternate method had a larger causal set size than MsCAVIAR, while points below the line indicate the opposite.

While the original loci totaled 29,479 SNPs, averaging 159.3 SNPs per locus, the causal sets returned by MsCAVIAR totaled 9,390 SNPs, averaging 50.8 SNPs per locus with a median of 31 SNPs. Meanwhile, PAINTOR's causal sets totaled 9,118 SNPs (49.3 average, 34 median), CAVIAR's sets using the UKB data totaled 11,538 SNPs (62.4 average, 44 median), and CAVIAR's sets using the BBJ data totaled 18,520 SNPs (100.0 average, 70 median). Thus, similarly to our simulation study's findings, MsCAVIAR and PAINTOR generally returned smaller causal set sizes than CAVIAR, and MsCAVIAR's median causal set size was slightly smaller than PAINTOR's. In contrast with the simulation study, MsCAVIAR's average causal set size was slightly larger than that of PAINTOR's. A full list of the loci we identified and the causal set sizes returned can be found in Table A in S1 Text.

As an additional way of viewing the results, we generated scatter plots of the causal set sizes at each locus for MsCAVIAR compared to those of PAINTOR and CAVIAR (Fig 4). This visualizes the comparative causal set sizes at individual loci. The scatter plots and their associated lines of equality reveal that MsCAVIAR's set sizes were consistently smaller than CAVIAR's across almost all loci, with one notable exception in which CAVIAR's causal set size was substantially smaller than MsCAVIAR's. The comparison with PAINTOR illustrates how MsCAVIAR's median causal set size was smaller than PAINTOR's but its average was higher: MsCAVIAR returned slightly smaller causal set sizes than PAINTOR for most loci, but in some cases, MsCAVIAR's causal set size was much larger than PAINTOR's, dragging MsCAVIAR's average causal set size above that of PAINTOR.

## Discussion

In this work, we introduced MsCAVIAR, a method for identifying causal variants in associated regions while leveraging information from multiple studies. Our approach requires only summary statistics as opposed to genotype data and handles heterogeneity of effect sizes, differing sample sizes, and different LD structures between studies, making trans-ethnic fine mapping an ideal application. We demonstrated that our method is well-calibrated and improves fine-mapping resolution in simulation studies. MsCAVIAR is available as free and open source software at https://github.com/nlapier2/MsCAVIAR.

We make several important assumptions in this model, which may not always be true. It has been shown that many causal SNPs are shared across populations [12, 17, 18]. MsCAVIAR is designed to leverage this phenomenon for increased power; however, causal variants may be

unique to one population. In those instances, MsCAVIAR's model doesn't match the data, so it may not be well-calibrated or it may return large causal sets. If one population has an obvious GWAS signal while the other population(s) lack even a marginally significant signal in the same locus, applying CAVIAR to the population with signal may be more appropriate.

We also assume that all studies are drawn with equal heterogeneity $\tau^2$. This is unlikely to be true if multiple studies are from a single population while another study is from a different population. In such a scenario, we recommend grouping the studies by population, running fixed effects meta-analysis on each group, and then running MsCAVIAR on the results for the different groups. Concretely, the input summary statistics for MsCAVIAR should be the results from the meta-analysis of each population, and the input LD matrices should be derived from either the genotype data (if available) or the appropriate reference panels for each population. However, it is still possible that even ostensibly different populations may be more similar to each other at certain loci than other populations. Therefore, we plan to extend our method to handle this case in future work.

In practice, we set the $\tau^2$ parameter to a fixed value, which was chosen to give power to detect both small and large amounts of heterogeneity (Methods, "Parameter Setting in Practice"). This value could, in principle, be adjusted based on the apparent heterogeneity present in the data. However, care would have to be taken to not overfit the parameter to the summary statistics in each locus, since the heterogeneity of different causal SNPs can vary across loci and some causal SNPs may be missed when the heterogeneity parameter is overfitted. Future work could develop a procedure for fitting this parameter.

Several methodological extensions to MsCAVIAR are possible as well. MsCAVIAR aims to return a causal set that contains all causal SNPs in a locus, while another fine mapping method, SuSiE [30] solves a complementary problem by returning one or more credible sets that each contain at least one causal SNP. The advantage of the former approach is its completeness in terms of identifying all causal signals, while the advantage of the latter approach is its ability to separate distinct causal signals within a locus into separate sets. A future extension to MsCAVIAR could aim to accomplish the benefits of both by returning a causal set with all causal SNPs, and then partitioning this set into distinct subsets with separate causal signals.

Functional information can in principle be factored into MsCAVIAR's model by modifying the prior distribution $P(C)$ so that not every variant has the same prior probability of being causal, as described in the CAVIAR paper [8]. However, setting these priors arbitrarily can yield misleading results, and future work is needed to determine how best to model various functional priors in the context of MsCAVIAR's model.

Finally, stochastic search could be used to speed up MsCAVIAR in cases where there are possibly many causal variants [10, 31]. MsCAVIAR's runtime is largely determined by the number of SNPs in the locus and the number of causal SNPs allowed: if there are $M$ total SNPs and up to $K$ are allowed to be causal, then there are potentially up to $\binom{M}{K}$ causal status vectors to evaluate. Thus, runtime can become an issue when there are many SNPs in a locus or many studies, and especially when users desire to allow for more than three possibly causal SNPs at a locus. Stochastic search can help reduce the search space by not evaluating every possible combination of causal SNPs, though this involves managing the risk of missing the optimally minimal causal set.

## Methods

### Overview of the MsCAVIAR model

In this section, we expand upon the high-level discussion of the method given in "MsCAVIAR Overview". We briefly describe the search for the minimal-sized causal set of SNPs and the

generative model behind it. In the following subsections, we describe the computational details in depth.

As discussed in the "MsCAVIAR Overview" section, our method takes as input the association statistics (i.e. Z-scores) and linkage disequilibrium (LD) matrix at the same locus for each study. MsCAVIAR computes and outputs a minimal-sized "causal set" of SNPs that, with probability at least $\rho$, contains *all* causal SNPs. By our definition of a causal set, every causal SNP must be contained in the set with high probability, but not every SNP in the set needs to be causal. Concretely, each SNP can be assigned a binary causal status: 1 for causal or 0 for non-causal. So long as none of the SNPs outside of the causal set are set to 1, the assignments are compatible with our definition of a causal set. We can represent these causal status assignments in a binary vector with one entry for each SNP denoting its causal status; we call such a vector a "configuration" and denote it as $C$. For a putative causal set of SNPs $\mathcal{K}$, we define $\mathcal{C}_{\mathcal{K}}$ as the set of causal configurations that are compatible with this causal set—the set of vectors with no '1' entries for SNPs not in $\mathcal{K}$.

For each configuration $C^*$ compatible with the causal set, we compute its (posterior) probability in a Bayesian manner:

$$P(C^*|S) = \frac{P(S|C^*)P(C^*)}{\sum_{C\in\mathcal{C}}P(S|C)P(C)} \tag{1}$$

where $S$ denotes the summary statistics for all input studies, and $\mathcal{C}$ is the space of possible causal status indicator vectors (including those not compatible with the causal set).

We must now define the likelihood $P(S|C)$ and the prior $P(C)$. For the prior, we assume that each variant is equally likely to be causal, with probability $\gamma$, and thus the prior probability $P(C)$ is

$$\prod_{j=1}^{m}\gamma^{C_j}(1-\gamma)^{1-C_j} \tag{2}$$

where $C_j$ is the $j^{th}$ entry (SNP) in $C$. The likelihood for $P(S|C)$ can be written as

$$S|C \sim \mathcal{N}(0, \Sigma + \Sigma\Sigma_C\Sigma) \tag{3}$$

where $\Sigma$ is a block-diagonal matrix where each block corresponds to one study's LD matrix and $\Sigma_C$ is a matrix modeling the covariance structure between the causal SNPs. Further computational details on the model are provided in the subsections below, but we will make two statements here for clarity.

The first being that the likelihood function in Eq 3 depends on the assumption that the summary data $S_q$ for each study $q$ is independently distributed as such: $S_q|\Lambda_q \sim \mathcal{N}(\Sigma_q\Lambda_q, \Sigma_q)$ where $\Sigma_q$ is the LD matrix for the study and $\Lambda_q$ is its non-centrality parameters. This is then coupled with the assumed prior for $\Lambda_q|C \sim \mathcal{N}(0, \Sigma_{C_q})$ where $\Sigma_{C_q}$ is the covariance structure between causal SNPs for study $q$. Using the distribution of $\Lambda_q$ as a conjugate prior, the overall distribution in a single study is $S_q|C \sim \mathcal{N}(0, \Sigma_q + \Sigma_q\Sigma_{C_q}\Sigma_q)$. This is restated and more fully described in the following section ("Fine mapping in a single study"), particularly in Eqs 8–11.

The second being that Eq 3 also depends on the assumption of how the causal variants in each study relate to one another. We began by concatenating the non-centrality parameters across Q studies where each contains M SNPs to create the QM-length vector $vec(\Lambda)$. The distribution of $vec(\Lambda)$ is $vec(\Lambda) \sim \mathcal{N}(0, \Sigma_c)$ where $\Sigma_c$ can be written using the following

Kronecker product ($\otimes$)

$$\Sigma_C = (\tau^2 I_Q + \sigma^2 1_Q 1_Q^T) \otimes \mathrm{diag}(1_{causal})_M \tag{4}$$

where $\sigma^2$ corresponds to the non-centrality parameter's variance as seen in the per-study setting (see Eq 10) which we assume is identical across studies and where $\tau^2$ captures the heterogeneity between studies (see Eq 21). Let $1_Q 1_Q^T$ be a matrix of all 1s, $I_Q$ an identity matrix, and $diag(1_{causal})_M$ be an ($M \times M$) diagonal matrix whose diagonal entries are given by the ($1 \times M$) indicator vector $1_{causal}$ whose entries $m$ are 1 if SNP $m$ is causal and 0 otherwise. The Kronecker product for $\Sigma_C$ is stated again as Eq 26 in the section "Efficient meta-analysis" where it is more fully described.

With these clarifications on the likelihood function $P(S|C)$, we now proceed to how we calculate the posterior probability that $\mathcal{K}$ contains all the causal SNPs:

$$P(\mathcal{C}_{\mathcal{K}}|S) = \sum_{C^* \epsilon \mathcal{C}_{\mathcal{K}}} P(C^*|S) \tag{5}$$

The goal is then to find the minimum-sized set $\mathcal{K}^*$ that has a posterior probability of at least $\rho^*$, called the "$\rho^*$ confidence set":

$$P(\mathcal{C}_{\mathcal{K}^*}|S) \geq \rho^* \tag{6}$$

This is done by evaluating causal configuration vectors with only one non-zero element, and then those with two non-zero elements, and so on until the end condition above is met. In practice, we limit the search space $\mathcal{C}$ by allowing the user to set the maximum number of causal SNPs allowed to 3 by default.

As stated previously, the following subsections explain the derivation of $S|C \sim \mathcal{N}(0, \Sigma + \Sigma\Sigma_C\Sigma)$, the structure of $\Sigma_C$, and computational efficiency details. We begin by reviewing fine mapping in a single study, and then proceed to the multiple study case.

## Fine mapping in a single study

We now describe a standard approach for fine mapping significant variants from a genome-wide association study (GWAS). In the GWAS, let there be $N$ individuals, all of whom have been genotyped at $M$ variants. For each individual $n$, we measure a quantitative trait $y_n$, resulting in the $N \times 1$ column vector $Y$ of phenotypic values. We denote $G$ as the $N \times M$ matrix of the genotypes where $g_{nm} \in \{0, 1, 2\}$ is the minor allele count for the $n$th individual at variant $m$. We standardize $G$ according to the population proportion $p$ of the minor allele and denote this as $X$ where $x_{ij} \in \{\frac{-2p}{\sqrt{2p(1-p)}}, \frac{1-2p}{\sqrt{2p(1-p)}}, \frac{2-2p}{\sqrt{2p(1-p)}}\}$.

We assume Fisher's polygenic model, which means $Y$ is normally distributed and each variant $x_m$ has a linear effect on $Y$. We, therefore, have the following model:

$$Y = \mu 1 + \sum_{m=1}^{M} \beta_m x_m + e \tag{7}$$

where $\beta_m$ is the effect size of variant $x_m$ and $e$ is the variation in $Y$ not explained by additive genetic effects and follows the Gaussian distribution $e \sim \mathcal{N}(0, \sigma_e^2 I)$.

We now model the observed summary statistics $S = [s_1, \ldots, s_m]$ according to

$$S|\Lambda_C \sim \mathcal{N}(\Sigma\Lambda_C, \Sigma) \tag{8}$$

where $\Sigma$ represents the pairwise Pearson correlations between the genotypes. $\Lambda_C = [\lambda_{C_1} \ldots \lambda_{C_M}]$ represents the true standardized causal effect sizes of each SNP, where each entry $\lambda_{C_m} = 0$ if SNP $m$ is non-causal and $\lambda_{C_m} \neq 0$ otherwise.

The distribution of $\Lambda_C$ can be defined as:

$$\Lambda_C | C \sim \mathcal{N}(0, \Sigma_C) \tag{9}$$

where $C = \{0, 1\}^M$ is an $M \times 1$ binary vector indicating whether each variant is causal, and

$$\Sigma_C = \begin{cases} 0, & \text{if } i \neq j. \\ \sigma^2, & \text{if } i \text{ is causal.} \\ \epsilon, & \text{if } i \text{ is not causal.} \end{cases} \tag{10}$$

and where $\epsilon$ is a small constant to ensure that the matrix $\Sigma_C$ is full rank. (We later relax the need for $\Sigma_C$ to be full rank in "Handling Low Rank LD Matrices"). Here, and below, we use the shorthand $\sigma^2$ to represent the variance of the $\lambda_{C_m}$ (see the subsection "Extending MsCA-VIAR to different sample sizes" for details on this parameter). The off-diagonals of $\Sigma_C$ are zero because the effect sizes of causal variants are independent of one another.

We use the shorthand $\Lambda = \Sigma\Lambda_C$ to refer to the non-centrality parameters (NCPs) of the statistics of all SNPs, which are induced by Linkage Disequilibrium (LD) with the causal SNPs. Thus, $S|\Lambda \sim \mathcal{N}(\Lambda, \Sigma)$. Since $\Lambda = \Sigma\Lambda_C$ and LD structure is symmetric ($\Sigma = \Sigma^T$), we have the following distribution for $\Lambda|C$:

$$(\Lambda | C) \sim \mathcal{N}(0, \Sigma\Sigma_C\Sigma) \tag{11}$$

We will now define $\gamma$ as the probability of a variant being causal, which makes the causal status for the $m$th variant a Bernoulli random variable with the following probability mass function: $f(c_m; \gamma) = \gamma^{c_m}(1 - \gamma)^{1-c_m}$. We assume the causal status for each variant is independent of the other variants, leading to the following prior for the our indicator vector: $P(C) = \prod_{m=1}^{M} \gamma^{C_m}(1 - \gamma)^{1-C_m}$. Assuming that each variant has a probability $\gamma$ of having a causal effect, the prior can then be written as follows:

$$P(\Lambda, C) = P(\Lambda|C)P(C) = f(\Lambda, 0, \Sigma_C)\prod_{m=1}^{M}\gamma^{C_m}(1 - \gamma)^{1-C_m} \tag{12}$$

where $f(\Lambda, 0, \Sigma_C)$ is the probability density function shown in Eq 11.

We determine which variants are causal by calculating the posterior probability of each configuration $C^* \in \mathcal{C}$, where $\mathcal{C}$ is the set of all possible configurations, given the set of summary statistics:

$$P(C^*|S) = \frac{P(S|C^*)P(C^*)}{\sum_{c \in \mathcal{C}} P(S|c)P(c)} = \frac{\int_{\Lambda_{C^*}} P(S|\Lambda, C^*)P(\Lambda = \Sigma\Lambda_{C^*}, C^*)d\Lambda_{C^*}}{\sum_{c \in \mathcal{C}} \int_{\Lambda_c} P(S|\Lambda, c)P(\Lambda = \Sigma\Lambda_c, c)d\Lambda_c} \tag{13}$$

For us to calculate the posterior probability of $C^*$ given $S$, we need to integrate over all possible values for the non-centrality parameters of the causal variants in $\Lambda$ in order to get the values of $\Lambda$ that makes observing $S$ most probable.

## Efficient computation of likelihood functions

The integral above is intractable in the absence of parametric assumptions about the data. Fortunately, a closed-form solution is available due to the fact that, when a conjugate prior is multivariate normally distributed, its predictive distribution is also multivariate normal. As shown above, $S|\Lambda \sim \mathcal{N}(\Lambda, \Sigma)$ and $(\Lambda|C) \sim \mathcal{N}(0, \Sigma\Sigma_C\Sigma)$. The predictive form of $S$ is then

$$S \sim \mathcal{N}(0, \Sigma + \Sigma\Sigma_C\Sigma) \tag{14}$$

However, computing the likelihood of $S$ with this distribution is still computationally expensive. Consider the multivariate normal probability density function, assuming the variable $Z$ below is MVN distributed with mean $\mu$ and covariance matrix $\Sigma$:

$$f(Z; \mu, \Sigma) = \frac{1}{\sqrt{(2\pi)^M |\Sigma|}} exp(-\frac{1}{2}(Z - \mu)^T \Sigma^{-1}(Z - \mu)) \tag{15}$$

For $S$, the covariance matrix is $\Sigma + \Sigma\Sigma_C\Sigma$, which has dimension $(M \times M)$, where $M$ is the number of SNPs in each study. Taking the determinant or inverse of this covariance matrix, as required by the above likelihood function, would take $O(M^3)$ time. Here, we demonstrate how to compute this likelihood efficiently, leveraging insights from several studies that have explored this topic [9, 10, 32].

We need to compute $S^T(\Sigma + \Sigma\Sigma_C\Sigma)^{-1}S$ and $|\Sigma + \Sigma\Sigma_C\Sigma|$ (note that our $\mu$ is 0). We can factor out $\Sigma$ from both of the equations above:

$$S^T(\Sigma + \Sigma\Sigma_C\Sigma)^{-1}S = S^T\Sigma^{-1}(I + \Sigma_C\Sigma)^{-1}S \tag{16}$$

$$|\Sigma + \Sigma\Sigma_C\Sigma| = |\Sigma||I + \Sigma_C\Sigma| \tag{17}$$

Notably, $S^T\Sigma^{-1}$ and $|\Sigma|$ can be computed once and re-used for every causal configuration $\Sigma_C$. Below, we assume $\Sigma$ is of full-rank; Lozano et. al [32] show how to address the low-rank case.

We use the Woodbury matrix identity [33], below, to speed up the matrix inversion equation:

$$(A + UEV)^{-1} = A^{-1} - A^{-1}U(E^{-1} + VA^{-1}U)^{-1}VA^{-1} \tag{18}$$

Here, we set $A = I_{M\times M}$, $E = I_{K\times K}$ where $K$ is the number of causal SNPs per study, and $UV = \Sigma_C\Sigma$. In particular, $U$ is the $(M \times K)$ matrix of rows corresponding to causal SNPs in $\Sigma_C$. We are taking advantage of the fact that rows corresponding to non-causal SNPs are zeros and thus do not affect the matrix multiplication. Similarly, $V$ is the corresponding columns of $\Sigma$, and is $(K \times M)$. Applying the Woodbury matrix identity to our case, we get:

$$\begin{aligned}(I_{M\times M} + \Sigma_C\Sigma)^{-1} &= (I_{M\times M} + UV)^{-1} \\ &= I_{M\times M}^{-1} - I_{M\times M}^{-1}U(I_{K\times K}^{-1} + VI_{K\times K}^{-1}U)^{-1}VI_{M\times M} \\ &= I_{M\times M} - U(I_{K\times K} + VU)^{-1}V\end{aligned} \tag{19}$$

Crucially, we are now inverting a $(K \times K)$ matrix instead of an $(M \times M)$ matrix, where $K \ll M$ since most SNPs are not causal [32]. We use Sylvester's determinant identity [34] to speed up the determinant computation as follows:

$$|I_{M \times M} + UV| = |I_{K \times K} + VU| \qquad (20)$$

Similarly, we are computing the determinant of a $(K \times K)$ matrix instead of an $(M \times M)$ matrix. Using these speedups, the computation of the likelihood function of $S$ is reduced from $O(M^3)$ to $O(K^3)$ plus some $O(MK^2)$ matrix multiplication operations, which is tractable under the reasonable assumption that each locus has at most $K = 3$ causal SNPs. In the "Efficient meta-analysis" subsection below, we discuss the computational complexity and the use of these efficient matrix computations in the multiple study setting.

## Fine mapping across multiple studies

As GWAS continue to grow in size, frequency, and diversity, there is an increasing need for fine mapping methods that leverage results from multiple studies of the same trait. A simple approach is to assume that there is one true non-centrality parameter for every variant; therefore $\Lambda_C$ is identical across studies. This approach is referred to as a fixed effects model. In this case, the $q$th study's $\Lambda_{C_q} = \Lambda_C$.

While there is evidence that many causal SNPs are shared across populations [12, 17, 18], the assumption that the true causal non-centrality vector $\Lambda_C$ is the same across studies is unrealistic, especially when the studies are measured in different ethnic groups [13, 17, 18].

We relax this assumption by utilizing a random effects model, in which each study $q$ is allowed to have a different $\Lambda_{Cq}$. Under this model, a causal SNP $m$ has an overall mean non-centrality parameter, which we denote with the scalar $\lambda_{C_m}$, from which the non-centrality parameter for SNP $m$ in each study $q$, denoted by the scalar $\lambda_{C_{mq}}$, is drawn with heterogeneity (variance) $\tau^2$. According to the polygenic model, $\lambda_{C_m}$ is distributed as $\lambda_{C_m} \sim \mathcal{N}(0, \sigma^2)$; therefore, $\lambda_{C_{mq}}$ is distributed as $\lambda_{C_{mq}} \sim \mathcal{N}(\lambda_{C_m}, \tau^2)$. Consequently, the vector $\Lambda_{Cm}$ for this SNP across all studies will have the following distribution:

$$\Lambda_{Cm} \sim \mathcal{N}(0, \sigma^2 11^T + \tau^2 I) \qquad (21)$$

where $Q$ is the number of studies, $1$ is a $(Q \times Q)$ matrix of 1s, and $I$ is the $(Q \times Q)$ identity matrix. Intuitively, since the SNP $m$ was drawn with variance $\sigma^2$, this variance component is shared across studies, while the variance component $\tau^2$ is study-specific and therefore it is only present along the diagonal of the covariance matrix. If a variant is not causal, its true effect size should be zero. We construct a matrix $\Lambda_C$ of size $(MQ \times MQ)$, where $M$ is the number of SNPs and each row corresponds to the $Q$-length vector $\Lambda_{Cm}$ corresponding to SNP $m$. In practice, we ensure that this matrix is full-rank by drawing the non-causal SNPs according to $\Lambda_{Cm} \sim \mathcal{N}(0, \epsilon I)$, where $\epsilon$ is a small constant.

From this we will now build out the posterior probability of $P(C^*|S_q)$ similarly to Eq 13. Now instead of $\Lambda_{C_q} = \Sigma_q \Lambda_C$ for study $q$, we have to account for $\Lambda_q = \Sigma_q \Lambda_{C_q}$ where $\Lambda_{C_q}$ is drawn from a multivariate normal distribution. This means we have to integrate over the

domain-space of $\Lambda_{C_q}$ to as well as $\Lambda_C$ to describe $P(C^*|S_q) = \frac{P(S_q|C^*)P(C^*)}{\sum_{C \in C} P(S_q|C)P(C)}$

$$P(C^*|S_q) = \frac{\int_{\Lambda_{C_q^*}} P(S_q|\Lambda_q, C^*) \int_{\Lambda_{C^*}} P(\Lambda_q = \Sigma_q \Lambda_{C_q^*}|\Lambda_{C^*}, C^*)P(\Lambda_{C^*}, C^*)d\Lambda_{C^*}d\Lambda_{C_q^*}}{\sum_{c \in C} P(S_q|\Lambda_q, c) \int_{\Lambda_{c_q}} P(S_q|\Lambda_q, c) \int_{\Lambda_c} P(\Lambda_q = \Sigma_q \Lambda_{c_q}|\Lambda_c, c)P(\Lambda_c, c)d\Lambda_c d\Lambda_{c_q}} \qquad (22)$$

## Efficient meta-analysis

Now that we have described the distribution of each SNP in our meta-analysis, we show how to jointly analyze them. We begin by explicitly defining the structure of the covariance matrix between studies by way of a small example with three SNPs at a locus in two different studies. Since the covariance of a matrix is undefined, we denote $vec(\Lambda_C)$ as the vectorized form of the original matrix ($\Lambda_C$). Concretely:

$$\mathrm{vec}(\Lambda_C) = \mathrm{vec}\left( \begin{bmatrix} \lambda_{C_{11}} & \lambda_{C_{21}} \\ \lambda_{C_{12}} & \lambda_{C_{22}} \\ \lambda_{C_{13}} & \lambda_{C_{23}} \end{bmatrix} \right) = \begin{bmatrix} \lambda_{C_{11}} \\ \lambda_{C_{12}} \\ \lambda_{C_{13}} \\ \lambda_{C_{21}} \\ \lambda_{C_{22}} \\ \lambda_{C_{23}} \end{bmatrix} \qquad (23)$$

Assume SNPs 1 and 3 are causal and SNP 2 is not causal. Then the vectorized form of the non-centrality parameters given the causal statuses has the following multivariate normal distribution:

$$(\mathrm{vec}(\Lambda_C)|\mathrm{vec}(C)) \sim \mathcal{N}\left( \begin{bmatrix} 0 \\ 0 \\ 0 \\ 0 \\ 0 \\ 0 \end{bmatrix}, \left[ \begin{array}{ccc|ccc} \sigma^2 + \tau^2 & 0 & 0 & \sigma^2 & 0 & 0 \\ 0 & \epsilon & 0 & 0 & 0 & 0 \\ 0 & 0 & \sigma^2 + \tau^2 & 0 & 0 & \sigma^2 \\ \hline \sigma^2 & 0 & 0 & \sigma^2 + \tau^2 & 0 & 0 \\ 0 & 0 & 0 & 0 & \epsilon & 0 \\ 0 & 0 & \sigma^2 & 0 & 0 & \sigma^2 + \tau^2 \end{array} \right] \right) \qquad (24)$$

We call the covariance matrix above $\Sigma_C$. Viewing $\Sigma_C$ as having a block structure, the blocks along the diagonal represent SNPs from the same study, while off-diagonal blocks represent SNPs from different studies. Here $\Sigma_C$ is $(3 * 2 \times 3 * 2) = (6 \times 6)$; in general, for $M$ SNPs and $Q$ studies, $\Sigma_C$ will be $(MQ \times MQ)$. In other words, there will be an $(Q \times Q)$ grid of $(M \times M)$ blocks. Within each block, the diagonal represents each SNP's variance, while the off-diagonal represents covariation between different SNPs. As SNPs are assumed to be independent, these are always 0. There are two variance components: the global genetic variance $\sigma^2$ from which the global mean non-centrality parameter for a SNP is drawn, and the heterogeneity between studies $\tau^2$. When a SNP is causal, its variance (its covariance with itself in the same study) will

contain both variance components ($\tau^2 + \sigma^2$), while its covariance with the same SNP in a different study will be $\sigma^2$, because they were drawn from the same overall non-centrality parameter with variance $\sigma^2$ but were drawn separately with variance $\tau^2$.

The $\Sigma_C$ above, leaving aside $\epsilon$ for now, can alternately be written in the more-compact form

$$\Sigma_C = \begin{bmatrix} \tau^2 + \sigma^2 & \sigma^2 \\ \sigma^2 & \tau^2 + \sigma^2 \end{bmatrix} \otimes \begin{bmatrix} 1 & 0 & 0 \\ 0 & 0 & 0 \\ 0 & 0 & 1 \end{bmatrix} \tag{25}$$

where $\otimes$ represents the Kronecker product operator. This can be further condensed and generalized into:

$$\Sigma_C = (\tau^2 I_Q + \sigma^2 1_Q 1_Q^T) \otimes \text{diag}(1_{causal})_M \tag{26}$$

where $Q$ is the number of studies, $M$ is the number of SNPs, $1_Q 1_Q^T$ is the ($Q \times Q$) matrix of all 1s, $I_Q$ is the ($Q \times Q$) identity matrix, and $diag(1_{causal})_M$ is an ($M \times M$) diagonal matrix whose diagonal entries are given by the ($1 \times M$) indicator vector $1_{causal}$ whose entries $m$ are 1 if SNP $m$ is causal and 0 otherwise.

As with CAVIAR, the $\epsilon$ entries along the diagonal are small numbers to ensure full rank. Also note that the CAVIAR model is a specific case of this model, in which there is only one study and thus there is no $\tau^2$ component. The CAVIAR $\Sigma_C$ has the same structure as the upper left block in the $\Sigma_C$ above, when there are 3 SNPs and $\tau^2$ is set to 0.

The efficient computation properties for the single-study case also apply to the multiple-study case. In the latter setting, the matrices that need to be inverted are ($MQ \times MQ$) instead of ($M \times M$), where $M$ and $Q$ are the number of SNPs in a locus and the number of studies, respectively. Consequently, in the Woodbury matrix identity equations, $U$ and $V$ are ($MQ \times KQ$) and ($KQ \times MQ$), respectively, where $K \ll M$ is the number of causal SNPs, and the matrix given by the Woodbury identity is ($KQ \times KQ$). Sylvester's determinant identity gives a matrix of this size as well. The computation time is thus reduced from $O(M^3 Q^3)$ to $O(K^3 Q^3)$.

## Handling low rank LD matrices

The methods described above assume that the Linkage Disequilibrium (LD) matrix is full rank, in order to invert this matrix in the process of computing the Multivariate Normal (MVN) likelihood function. In practice, this is often not the case, because SNPs are sometimes in perfect LD. This can even happen when SNPs are not in perfect LD due to many highly correlated SNPs being a linear function of each other. CAVIAR [8] employs a method to add a small amount of random noise to the diagonal of the LD matrix to avoid this, but we found this adjustment to be insufficient to avoid the latter situation when LD matrices were sufficiently large, especially with blocks of high-LD.

Lozano et al [32] developed a method for computing the MVN likelihood function when the LD matrix is low rank. MsCAVIAR implements this method and thereby avoids the aforementioned low rank issue. We briefly describe the method below on an intuitive level, but readers should refer to the work by Lozano et al [32] for the full derivation.

Since the LD matrix $\Sigma$ is positive semi-definite, it can be eigendecomposed as follows:

$$\Sigma = W \Omega W^T \tag{27}$$

where $W$ is the matrix of eigenvectors, such that the $i$-th column of $W$ is the $i$-th eigenvector of $\Sigma$, and $\Omega$ is a diagonal matrix that consists of eigenvalues of $\Sigma$ where the $i$-th diagonal element

of $\Sigma$ is the $i$-th eigenvalue of $\Sigma$. Lozano et al. then introduce a new set of summary statistics $S' = \Omega^{-1/2} W^T S$ which, using some algebra, is shown to have the joint distribution

$$S' = \Omega^{-1/2} W^T S \sim \mathcal{N}(0, I + mB\Sigma_C B^T) \tag{28}$$

where $I$ is the identity matrix, $m$ is the number of SNPs, and $B = \Omega^{-1/2} W^T$. Since $I + mB\Sigma_C B^T$ is full rank, we can compute the likelihood function for $S'$, even when $S$ is not full rank.

In order to evaluate the likelihood function for our original summary statistics $S$, we first transform the original summary statistics $S$ to $S'$ via $S' = \Omega^{-1/2} W^T S$, and then apply the above procedure to evaluate the likelihood function for $S'$. This obviates the need for the $\epsilon$ parameter previously used to ensure full rank in the definition of $\Sigma_C$, so we now define $\Sigma_C$ in the single study setting as

$$\Sigma_C = \begin{cases} 0, & \text{if } i \neq j \text{ or SNP } i \text{ is not causal.} \\ \sigma^2, & \text{if SNP } i \text{ is causal.} \end{cases} \tag{29}$$

## Extending MsCAVIAR to different sample sizes

In "Fine mapping across multiple studies", we discussed the MsCAVIAR model, in which the non-centrality parameters $\lambda_{C_{mq}}$ for SNP $m$ in each study $q$ are drawn around a global mean non-centrality parameter $\lambda_{C_m} \sim \mathcal{N}(0, \sigma^2)$ with variance $\tau^2$, such that $\lambda_{C_{mq}} \sim \mathcal{N}(\lambda_{C_m}, \tau^2)$. We note that $\lambda_{C_m}$ is itself a function of the non-standardized effect size $\beta_m$, where $\lambda_{C_m} = \frac{\beta_m \sqrt{N}}{\sigma_e}$ and $\beta_m \sim \mathcal{N}(0, \sigma_g^2)$. Thus, $\lambda_{C_m}$ and its variance $\sigma$ are functions of the sample size $N$. Since the sample size may not be consistent across the studies, this $\lambda_{C_m}$ is an oversimplification that cannot be used when different studies have different sample sizes. Below, we show how to model the $\lambda_{C_{mq}}$ for each study while taking into account possibly different sample sizes.

We will again draw the $q$th study's non-centrality parameter for variant $m$ according to this model. Each study $q$ has its own sample size $N_q$ and environmental component $\sigma_{e_q}$, and we draw it with heterogeneity parameter $\tau^2$ as previously defined, so

$$\lambda_{C_{mq}} \sim \mathcal{N}(\frac{\beta_m}{\sigma_{e_q}} \sqrt{N_q}, \tau^2) \tag{30}$$

We will now operate under the standard assumption that the trait has unit variance and variance explained by any particular SNP is small, thus $\sigma_e \approx 1$.

$$\Sigma = W\Omega W^T \tag{31}$$

Using our previous definition for a single study, we now have

$$\Lambda | C \sim \mathcal{N}(0, \Sigma_C) \tag{32}$$

where

$$\Sigma_C = \begin{cases} 0, & \text{if } i \neq j \text{ or SNP } i \text{ is not causal.} \\ \sigma^2, & \text{if SNP } i \text{ is causal.} \end{cases} \tag{33}$$

We now define $\sigma^2$ more formally to be $\sigma_g^2 N_q$ for the $q$th study, in the single study setting. In the multiple study setting, when we consider our matrix

$$\Sigma_C = \begin{bmatrix} \tau^2 + \sigma^2 & \sigma^2 \\ \sigma^2 & \tau^2 + \sigma^2 \end{bmatrix} \otimes \begin{bmatrix} 1 & 0 & 0 \\ 0 & 0 & 0 \\ 0 & 0 & 1 \end{bmatrix} \tag{34}$$

the $\sigma^2$ along the diagonal is defined identically to the precise single study definition; however, when modeling multiple studies, this adjustment changes the covariance between causal variant for two studies. We now define $\sigma^2 = \sqrt{N_{q1}}\sqrt{N_{q2}}\sigma_g^2$ for two studies $q1$ and $q2$ with population sizes $N_{q1}$ and $N_{q2}$. Note that if two studies have the same population size $N$, we get the original definition of $\sigma^2 = \sqrt{N}\sqrt{N}\sigma_g^2 = N\sigma_g^2$.

## Parameter setting in practice

Traditionally, the effect size $\beta \sim \mathcal{N}(0, \sigma_g^2)$ would be derived as a notion of the per-snp heritability. Here we do not define $\sigma_g^2$ as such, but rather treat it as an abstraction: we avoid making any assumptions on how heritable the given trait is and how that heritability is partitioned between loci. The way we set this parameter in practice is as a parameter for statistical power. If study $q1$ has the smallest sample size, we set this value such that $\sigma = \sigma_g^2 N_{q1} = 5.2$ for all variants. This value corresponds to the traditional genome-wide significant Z-score of 5.2, for which the two-sided Wald test p-value is $5 \times 10^{-8}$, which is considered significant by (conservatively) correcting for multiple testing [35]. Then the NCP for variant $m$ in the corresponding study $q1$ is $\lambda_{C_{q1,m}} \sim \mathcal{N}(5.2, \tau^2)$. For another study $q2$ with larger sample size, its NCP is drawn as $\lambda_{C_{q2,m}} \sim \mathcal{N}(5.2\sqrt{\frac{N_{q2}}{N_{q1}}}, \tau^2)$.

This value of $\sigma_g^2$ may not represent the actual heritability partitioning, but we set the parameter this way in our method for the practical purpose of giving MsCAVIAR power to fine map borderline significant variants in the smallest study. Similarly, we set $\tau^2 = 0.52$ by default, e.g. 10% of the value of $\sigma = \sigma_g^2 N_{q1}$, with the value chosen to give power to detect both small and large amounts of heterogeneity. We empirically observed that small misspecifications in the heterogeneity parameter do not have a substantial adverse effect (Fig G in S1 Text).

## Supporting information

**S1 Text. Additional simulations and a table of the real data results.**
(PDF)

## Author Contributions

**Conceptualization:** Nathan LaPierre, Kodi Taraszka, Farhad Hormozdiari, Eleazar Eskin.

**Formal analysis:** Nathan LaPierre, Kodi Taraszka, Helen Huang, Rosemary He.

**Funding acquisition:** Eleazar Eskin.

**Investigation:** Nathan LaPierre, Kodi Taraszka, Eleazar Eskin.

**Methodology:** Nathan LaPierre, Kodi Taraszka, Farhad Hormozdiari, Eleazar Eskin.

**Software:** Nathan LaPierre, Kodi Taraszka, Helen Huang, Rosemary He.

**Supervision:** Eleazar Eskin.

**Visualization:** Nathan LaPierre, Kodi Taraszka, Helen Huang, Eleazar Eskin.

**Writing – original draft:** Nathan LaPierre, Kodi Taraszka, Helen Huang, Eleazar Eskin.

**Writing – review & editing:** Nathan LaPierre, Kodi Taraszka, Eleazar Eskin.

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
