## [Decision Letter · Decision Letter 0]

5 Oct 2020

Dear Dr Eskin,

Thank you very much for submitting your Research Article entitled 'Identifying Causal Variants by Fine Mapping Across Multiple Studies' to PLOS Genetics. Your manuscript was fully evaluated at the editorial level and by independent peer reviewers. The reviewers appreciated the attention to an important problem, but raised some substantial concerns about the current manuscript. Based on the reviews, we will not be able to accept this version of the manuscript, but we would be willing to review again a much-revised version. We cannot, of course, promise publication at that time.

If you decide to revise the manuscript for further consideration at PLOS Genetics, please aim to resubmit within the next 60 days, unless it will take extra time to address the concerns of the reviewers, in which case we would appreciate an expected resubmission date by email to plosgenetics@plos.org.

You can use the link below to log into the system when you are ready to submit a revised version, having first consulted our Submission Checklist.

[LINK]

We are sorry that we cannot be more positive about your manuscript at this stage. Please do not hesitate to contact us if you have any concerns or questions.

Yours sincerely,

Eleftheria Zeggini

Associate Editor

PLOS Genetics

David Balding

Section Editor: Methods

PLOS Genetics

Reviewer's Responses to Questions

**Comments to the Authors:**

Reviewer #1: Summary: Minority population GWAS and trans-ethnic finemapping have increased in popularity due to the diversity in LD patterns across populations and how this diversity can be exploited to improve fine-mapping resolution. To this end, the authors present the statistical finemapping method “MsCAVIAR”, which is an extension of the previously developed Bayesian fine-mapping software CAVIAR. The primary advantage and novelty of the proposed approach is that it simultaneously addresses both effect heterogeneity and multiple causal variants at a given locus of interest for trans-ethnic finemapping. The authors’ perform a number of simulation studies as well as real data application to demonstrate the performance of their approach against leading methods that accommodate multiple causal variants. The method performs as well or better than its primary competitor (PAINTOR) in many regards. Overall, the paper is clear and well-written. The authors’ carefully outline their statistical methodology, addressing a number of computational hurdles. They address a number of limitations in their assumptions and highlight the strengths/weaknesses of their method. The authors also make code publicly available on Github, which appears to be complete and well-documented. I do have some questions regarding computational efficiency, and there are some issues with respect to consistency of notation – although most of my comments are minor.

As the authors’ note, there are a few limitations on their assumptions regarding tau and how it is fixed a priori in application. This corresponds both across loci and across studies, where imbalance in contributing studies vis a vis ancestral population may be a concern (as published GWAS are in predominantly European populations). Regarding the latter, one advantage of MR-MEGA in trans-ethnic finemapping is how it leverages meta-regression to account for distribution of genetic ancestry across the contributing individual studies, thus decomposing the effect heterogeneity into ancestral and random components. To this end, the authors suggest selecting one study from each population for inclusion. However, this naturally reduces finemapping efficiency depending on the distribution of study sizes. Would an alternative strategy using MsCAVIAR be to initially conduct fixed-effects meta-analyses within homogenous populations, where warranted, and then combine diverse population results for trans-ethnic finemapping?

Speaking of which – the authors initially mention MR-MEGA in the introduction but do not discuss it any further or include it in their comparisons. Given that the MR-MEGA paper demonstrates improved finemapping over PAINTOR in their paper, I assume the rationale for its exclusion for comparative performance analyses is due to its limiting assumption regarding number of underlying causal variants?

How would the authors additionally integrate functional annotation into their fine-mapping method, similar to PAINTOR?

It may be useful to quickly mention in “Parameter setting in practice” the relevance of 5.2 as a Z-score in relation to the traditional genome-wide significance criterion (i.e., the corresponding p under two-sided Wald test would be ~5-e08). Similarly - I’m not finding any clear justification for MsCAVIAR’s default value of tau2 = 0.52 – I assume the connection lies with just taking forcing a mean/variance relationship in effect heterogeneity for Z-scores at the genome-wide significance threshold?

Equation 10. It’s kind of confusing to use T to denote both the number of studies as well as the transpose operator. Similarly, later in the methods n is used to denote the number of studies, but also the number of subjects within a study. And then again it seems that ‘m’ is used to index study in “Extending MsCAVIAR to different sample sizes”. Some care should be used in maintaining consistent notation.

It’s not immediately clear how the computational cost scales with respect to number of included studies. Given the single study setting is O(k^3 ) plus some O(mk^2) – do we replace k with (k*n) to get the computational costs, where n = # of studies? Thus, is # of studies a similarly strong limiting factor in computational burden as k, which would potentially motivate the population-wise study aggregation study mentioned above?

Reviewer #2: LaPierre and colleagues present a novel approach for fine-mapping loci using summary statistics from multiple studies whilst accounting for heterogeneity in the effects of causal variants between them. The methodology can allow for multiple causal variants at a locus, and can be applied in the context of trans-ethnic fine-mapping by allowing for study-specific patterns of LD between variants. The methodology tackles an important challenge in human genetics, is extremely timely, and likely to be of great interest to the readership of PLoS Genetics. I am looking forward to trying the software! The manuscript is generally well written, although some additional details of the simulation study and the applications to trans-ethnic GWAS of type 2 diabetes and cholesterol would be useful.

Comments on methodology

1. Presumably it would be straightforward to incorporate a non-uniform prior of causality? Given the availability of enrichment in associations in specific annotations, it would be really useful to allow this flexibility in the method/software.

2. In the context of trans-ethnic meta-analysis, could the authors provide some guidance as to whether each study should be included separately in the meta-analysis, or whether a fixed-effects meta-analysis of each ethnic group should undertaken first, and then used as input to msCAVIAR? Presumably this would be computationally more demanding, but are there advantages in allowing for heterogeneity between studies from the same ancestry?

3. Heterogeneity is modelled under a random-effects model, but would it be feasible (and beneficial) to consider alternative models (such as less heterogeneity between more genetically similar studies)? Could the tau2 parameter be considered as a hyperparameter to be estimated, and could this give some intuition as to the extent of heterogeneity?

4. It wasn’t totally clear to me whether the (maximum) number of causal variants needs to be specified in advance.

5. Some details of computational efficiency would be beneficial. In simulations and data applications, SNPs are thinned by various criteria. Is this because the methodology/software does not work well if there are SNPs in strong LD, or is computationally demanding if the number of SNPs is large?

6. One of the nice features about Susie is that it effectively gives a credible set for each causal variant (reflecting the fact that these each represent a distinct association signal). If we were then keen to colocalise association signals with eQTLs, this could be done for each credible set separately. However, for msCAVIAR, it does not seem that we could extract equivalent information. For example, if there were two signals at a locus (and two causal variants), the first signal might be easy to fine-map, so that we are clear of the causal variant for that signal, but for the other signal, there might be several variants with equivalent fine-mapping support. Is there anyway to distinguish the fact that the variants in the credible set are somehow grouped by distinct associations, and if not, do the authors view this as a disadvantage?

Comments on the simulation study

1. Details of the simulation study are rather scant. How large are the two regions (physical distance and number of SNPs)? I can understand removing SNPs in perfect LD to select causal variants (although in practice, I guess two causal variants could be in perfect LD), but then this only leaves 48/38 SNPs, which does not seem like a “realistic” fine-mapping scenario. I also was not clear if the SNPs included in the analysis were also LD pruned, or if all SNPs in the region were considered as potentially causal. Could there every be a situation where a causal variant was specific to just one population (i.e. monomorphic in the other) – or did causal variants have to be present in both populations (at some frequency)?

2. What is Sigma_i on line 118? It might be described later, but it is not clear what it is at this point in the manuscript.

3. The authors suggest that the improved performance of msCAVIAR over PAINTOR could be because of the modelling of heterogeneity. Could the authors investigate this further by simulating effects of causal SNPs that are homogenous across studies?

4. Line 151. The authors state that the fact that msCAVIAR performs better than CAVIAR applied to each population is an indication of the improved fine-mapping resolution offered by trans-ethnic meta-analysis. However, could this actually be a reflection of the larger total ample size used by msCAVIAR? Could the authors also run simulations where the sample size of the population-specific studies are the same as a trans-ethnic study (i.e. just simulate two European studies of equal size, and compare with one European and one East Asian study of equal size)?

Comments on data applications

1. Effective sample size is a more useful way of representing the sample size for a disease phenotype – would actually better to give the number of diabetes cases and controls for each study.

2. It would be good to give information on the loci used in comparisons in supplementary information (for both applications), together with the numbers of credible causal variants for each locus with the different methods.

3. Centering the loci 50kb up- and down-stream of the lead SNP seems rather restrictive – we know that LD often extends over greater distances, and I think using 500kb up- and down-stream would be much more realistic. Was this done for computational reasons?

4. I didn’t follow the motivation for removing SNPs with p>0.0001. There could be examples where a causal variant does not have strong association in a single SNP analysis, and is only revealed when considering multiple causal variants (depending on patterns of LD between causal SNPs and directions of effect on the outcome). Was this done for computational reasons?

5. Figures 4 and 5. I did not find the violin plots useful in Figure 4. I understand that it is hard to summarise results when there are just five loci in Figure 4. It would be useful to have three scatter plots where the x-axis was the credible set size in msCAVIAR and the y-axis was the credible set size in one of each of the three other methods (i.e. each point is a locus) – this will provide useful information on within locus comparisons that could not be assessed with the current presentation. The box and whisker plots in Figure 5 are more useful, but I think could also be presented alongside scatter plots as described above.

6. I wasn’t totally clear about the final sentence of the last paragraph (line 259). In particular, I wasn’t clear about why it mattered that msCAVIAR models heterogeneity as being the same at each locus. As far as I am aware, PAINTOR also models heterogeneity as being the same at each locus (i.e. fixed effects, so no heterogeneity). So do these results imply that msCAVIAR does not perform well if effects are homogenous across studies? I think this emphasizes the importance of running some simulations under a model in which effects are the same in the two studies.

Comments on the discussion

1. I think it would be beneficial to expand somewhat on the comment about “equal heterogeneity” – presumably this is just the underlying assumption of a random effects model? I agree that this model is not appropriate when several studies are from one population, and one study is from another (because less heterogeneity would be expected between studies from the same population). However, I disagree with the recommendation to use a single study from each population. It would be much better to meta-analyse studies from the same population/ethnicity together, and use those as input to msCAVIAR (assuming you can use the same LD matrix for all studies from the sample population).

Reviewer #3: Review of LaPierre et al

The paper presents an extension of the fine-mapping

method (CAVIAR) to deal with multiple studies, allowing

for heterogeneity in effects among studies.

The paper uses simulation to show that this extension

(MsCAVIAR) produces better localization, in that

it reduces the size of the "causal set" produced compared with CAVIAR

(and other methods) run on the individual studies.

Results on real data show similar trends (smaller causal sets.)

This is a potentially useful -- if conceptually fairly straightforward --

extension of the CAVIAR method. However, there are several important

issues that would need addressing to make it suitable for publication.

Main Issues

1. While the main text is very clearly written and nicely presented,

the Methods section is confusing and difficult to follow.

I suggest the following:

First, there needs to be a very simple and clear statement of the model and

prior distribution used. This should be separated from any "derivation" of this

model (which can likely be mostly justified by appropriate citations of previous work)

and also separated from the computational tricks (which also seem like

straightforward extensions of previous work).

At the moment the model is very hard to extract from the text.

Lambda is used in different places in different ways: at the

top of p11 lambda_i = beta_i sqrt(n_i)/sigma_e , but then

later in the same paragraph it is used for the mean of S, which is

not the same thing. Maybe because of this there appear to be

circular definitions (Lambda_C is defined in terms of Lambda at

(2), and then Lambda is defined as \\Sigma Lambda_C at the top

of p15). [I actually don't think you need both Lambda and Lamba_C:

you can just use Lamdba for the true non-centrality parameters (which will

be 0 for non-causal SNPs) and then directly use \\Sigma Lambda for

the expectation of S, so

S| Lambda \\sim N (\\Sigma Lambda, \\Sigma)

or something like this?]

The extension to different sample sizes is described

imprecisely in words (top of p23), and needs equations to make it precise.

I would suggest just giving the model for different sample sizes

directly, since the case where they are the same are then a special case.

The model seems to be a "matrix normal" model, and making that explicit could help.

Second, the definitions of the summary data S need to be made

clear. At the moment they are defined as beta-hat_i \\sqrt(n_i)/sigma_e

but sigma_e is unknown. And at line 444 you say "we now operate... that

sigma_e has been standardized (sigma_e=1)". But there is no

way to standardize to ensure sigma_e=1 because we do not know the true residual variance.

It is common to standardize $y$ to have unit variance, but this does not imply the residual

variance is 1. (I think maybe in the model you are assuming $y$ has been standardized

to have variance 1, and then making the approximation that sigma_e \\approx 1

under the assumption of low heritability? But not sure whether you are also

doing this for s_i, so taking s_i = beta-hat_i \\sqrt(n_i) ? Or using an estimate

of the residual variance? In any cae these kinds of details, assumptions

and approximations need to be more precise.)

2. The simulation study is rather too favorable to the method, and should

be made more realistic. In particular i) the simulation is performed under

the assumed summary data model, rather than under a more realistic

full data model (ie the regression model (1)); ii) the simulation

is performed with "effect sizes" (actually, non-centrality parameter)

with a narrow range centered on 5.2, which not only

seems also to be used in the prior, but also seems unrealistic - it will seldom

produce either small difficult-to-detect effects or very large effects,

both of which are likely to occur frequently in practice;

iii) the simulation is done assuming the same LD structure in

both the study and the inference, whereas the real data analysis

uses a panel to approximate the study LD.

It would seem easy to generate more realistic simulated data

by simulating outcomes Y from the full data model (1), using real genotype

data (X) an a range of beta values (eg randomly drawn from N(0,sigma^2_g)

for some sigma^2_g) so that both small and big effects occur.

3. As argued in Wang et al, the idea of outputting a "causal

set" that, with high probability, contains *all* causal SNPs is

flawed. There are two reasons for this. First, it can ignore a lot of useful information.

For example, suppose there are two causal SNPs, and that one is in LD with

just itself, but the other is in LD with 50 others. Then the causal set will

contain (at least) 52 SNPs, and does not include the information that one SNP is very

precisely mapped (which a user could clearly find helpful!)

Second, if we allow that SNPs may have small effects, which is realistic,

then it becomes impossible for any method to be confident to include all the causal SNPs in a set

(at least, not without the set being very large). The paper sidesteps this

issue by avoiding simulations with small effects, which should

be rectified (see 2 above).

In light of this it seems unsatisfactory to rely on the size of the causal set

as the only indicator of improved performance, and I think the paper should

also provide other evidence for the superiority of the multi-study approach.

One possibility would be to demonstrate the benefits in terms of PIPs (posterior

inclusion probabilities). For example, does MsCAVIAR show a better

precision-recall curve (equivalently, true-positive rate vs false discovery rate)

as PIP threshold is modified?

4. The filtering in the real data analysis seems very ad hoc. and it is not clear

why it is done or whether it is necessary. Is it necessary to make the method's

performance look good compared with other methods? If so, this seems worrying.

If not, why not present results on much less filtered data? (even if it

may be more computationally intensive).

To comment in more detail on the filters: i) Discarding SNPs with marginal

p values >0.0001 could miss signals as SNPs can become more significant

once one controls for other SNPs in LD. ii) The logic that if the peak SNP is genome-wide

significant in one population and >0.0001 in another then the second population

won't help with localization isn't clear: first,

the potentially different patterns of LD in the two

populations mean that a second population could still help with localization even

without a genome-wide significant association; second maybe there are secondary SNPs

that will only show up as significant when one analyzes both populations.

iii) You say "fine mapping is not as useful when there are few strongly associated SNPs".

Why? I would think these loci may give the potential to fine map quite precisely!

Surely the problem cases are whether there are many SNPs in strong LD, all strongly associated,

which makes fine mapping difficult?

iv) "As a final step, we pruned groups of SNPs that were perfectly correlated with

each other in both studies... would cause the LD matrix to be low rank". Does

this mean you pruned if they were perfectly correlated in the *study* samples

or in the LD panel? It could make sense to pool together SNPs that are perfectly

correlated in the study, but if they are perfectly correlated in the panel

but not in the study then it seems you would want to keep them both.

(In that case perhaps you need the methods that allow for

low rank LD matrices in the panel, eg using the methods you cite

from Lozano et al.)

Other issues

- the caption to Figure 2 should include some explanation of the fact that the

calibration of SuSie can't be compared to the other methods because its sets

have a different goal.

- at line 110-111, I understand you pruned SNPs in perfect LD to reduce computation

and possibly to reduce low-rank issues for MsCAVIAR. However, while pruning may initially

seem innocuous, it raises several concerns. For example, a group of 10 SNPs in complete LD should

have approximately 10 times the probability that at least one of them

is causal compared with a single SNP that is in LD only with itself.

Most analysis methods would take that into account if the SNPs were simply

in "very high LD", but this is hard to do if

you have pooled/pruned the SNPs. And the pruning will understate

the size of typical causal sets (and so overstate performance) for all methods.

Also posterior quantities of interest (eg posterior inclusion probabilities)

may be difficult to correct for this pooling. The bottom line: if

pooling is just a way to reduce computation, can you show

that results of analyzing data with pooling are similar to results

without pooling?

- line 131-3 suggest that Fig 2 boxplots are for only a subset of

the simulations (even though the Figure caption does not mention it).

That seems dangerous, and I do not see why

not to include all simulations in the boxplot.

- line 148-9; they are only equivalent under the *assumption* that there is 1 causal SNP,

and not when methods are applied to data where the truth is only 1 causal SNP but they

do not assume only 1 causal SNP.

- line 218: Do the reported set sizes include all SNPs that were pruned

for being in complete LD with

selected SNPS? It seems that they should in order to give an accurate impression

of the effectiveness of fine mapping in practice.

- In the real data, how many causal SNPs do you estimate/identify at each locus?

- l273: this advice seems useless because how would one know? How much worse is MsCAVIAR

than CAVIAR if effects are unique to one population? One might hope that it

would be robust to this because of the heterogeneity in the model, and

this robustness seems worth assessing in simulations.

- Please provide a stable link to the code used to perform the simulations and data analysis

Details:

- l99: "dividing by the sum of posterior probabilities of all configurations": isn't this necessarily 1?

- l101: "continue increasing the size of the causal set" how is this done?

- l107-8: do you mean 20%/80% of SNP *pairs*?

- l114: don't use effect sizes and non-centrality parameter interchangeably as they are different.

- l115: casual -> causal

- l153 effect size -> non-centrality parameter

- l190: Should "White European" be "White British"?

- In the methods section using (lower-case) sigma for a variance (eg in equation (4)

and subsequently) is confusing. Use sigma^2 for a variance. Related to this, line 294 I think should be

sigma_e^2, and line 448 \\sigma_g should be squared.

- l338 "the integral above is intractable..." but then you say closed form is available!

- l339,341 posterior predictive -> predictive

- l352: " rows ... are zero" isn't this only true if you set epsilon=0?

It would be helpful to be more consistent throughout about treatment

of epsilon and what value it takes.

- in equation before l438 sqrt(n_m) should be n_m?

- line 459, say where 5.2 comes from

**Have all data underlying the figures and results presented in the manuscript been provided?**

Reviewer #1: Yes

Reviewer #2: **No: **I don't think the specific details of the data in Figures 4 and 5 have been presented - I've made some suggestions in my comments about providing additional supplemental information to deal with this.

Reviewer #3: **No: **

PLOS authors have the option to publish the peer review history of their article (what does this mean?). If published, this will include your full peer review and any attached files.

Reviewer #1: No

Reviewer #2: No

Reviewer #3: **Yes: **Matthew Stephens

---

## [Decision Letter · Decision Letter 1]

11 Jun 2021

Dear Dr Eskin,

Thank you very much for submitting your Research Article entitled 'Identifying Causal Variants by Fine Mapping Across Multiple Studies' to PLOS Genetics.

The manuscript was fully evaluated at the editorial level and by independent peer reviewers. The reviewers appreciated the attention to an important topic but identified some concerns that we ask you address in a revised manuscript

We therefore ask you to modify the manuscript according to the review recommendations. Your revisions should address the specific points made by reviewer 3.

[LINK]

Yours sincerely,

Eleftheria Zeggini

Associate Editor

PLOS Genetics

David Balding

Section Editor: Methods

PLOS Genetics

Reviewer's Responses to Questions

**Comments to the Authors:**

Reviewer #1: The authors have largely addressed all of my concerns. I have no further comment.

Reviewer #2: The authors have made extensive revisions to the manuscript and have dealt with all comments. I look forward to trying out the software!

Reviewer #3: Reveiew of LaPierre et al (revised)

The authors have been very responsive to the previous reviews, and the manuscript is much improved.

I have just one major remaining issue, and a few remaining minor comments I would ask the authors to address.

Main issue:

1. Although the new simulation study is much improved, it still shows results only for "in sample" LD matrices.

As the authors note, when using summary data it is very common to use "out of sample" LD matrices (eg LD matrices from a similar population reference panel like 1000 Genomes), and indeed the real data

analysis used that out-of-sample strategy. It is important for readers and users to understand whether this impacts accuracy, and so the simulation study should show results for both in-sample and out-of-sample LD matrices.

Other issues:

2. The MSCaviar model is essentially specified bymthe equation after l296, S|C \\sim N(0, \\Sigma + \\Sigma \\Sigma_C \\Sigma). I suggest to number that equation, since it is key. Further, please add at this point the precise descriptions of \\Sigma and \\Sigma_C. \\Sigma is a block-diagonal matrix (not a diagonal matrix) with each block containing the LD matrix from a study. And \\Sigma_C is given in the equation after l429, which you could

give here. It might seem a minor point, but it took me quite a while to collect those pieces all together to convince myself the model had been fully specified. Putting them all in the same place will help the reader see the model immediately, and numbering the key equations would help future writers refer to them if necessary.

3. I believe it would be further useful to state here that this model comes from assuming S_q | \\Lambda_q \\sim N(\\Sigma_q Lambda_q, \\Sigma_q) independently for each q where S_q are the summary data in study q,

Sigma_q is the LD matrix in study q and Lambda_q is the NCP in study q. (This basic assumption is stated in words in the main text, but seems worth repeating here). This is combined with a prior on Lambda

vec(\\Lambda) \\sim N(0, \\Sigma_C) where vec(\\Lambda) is the KM vector of concatenated NCPs from the K different studies to give the final model.

4. In equation (10) I would assume S_q denotes the data for study q. But then this equation seems irrelevant as you want p(C* | S) [ie posterior given all the data] and not p(C* | S_q).

**Have all data underlying the figures and results presented in the manuscript been provided?**

Reviewer #1: Yes

Reviewer #2: Yes

Reviewer #3: None

PLOS authors have the option to publish the peer review history of their article (what does this mean?). If published, this will include your full peer review and any attached files.

Reviewer #1: No

Reviewer #2: **Yes: **Andrew Morris

Reviewer #3: No

---

## [Editor Report · Decision Letter 2]

21 Jul 2021

Dear Dr Eskin,

We are pleased to inform you that your manuscript entitled "Identifying Causal Variants by Fine Mapping Across Multiple Studies" has been editorially accepted for publication in PLOS Genetics. Congratulations!

Yours sincerely,

Eleftheria Zeggini

Associate Editor

PLOS Genetics

David Balding

Section Editor: Methods

PLOS Genetics

Comments from the reviewers (if applicable):

**Data Deposition**

http://datadryad.org/submit?journalID=pgenetics&manu=PGENETICS-D-20-01186R2

**Press Queries**

---

## [Editor Report · Acceptance letter]

3 Sep 2021

PGENETICS-D-20-01186R2 

Identifying Causal Variants by Fine Mapping Across Multiple Studies 

Dear Dr Eskin, 

We are pleased to inform you that your manuscript entitled "Identifying Causal Variants by Fine Mapping Across Multiple Studies" has been formally accepted for publication in PLOS Genetics! Your manuscript is now with our production department and you will be notified of the publication date in due course.

With kind regards,

Andrea Szabo

PLOS Genetics

On behalf of:
